# The Influence of the Mixed DPC:SDS Micelle on the Structure and Oligomerization Process of the Human Cystatin C

**DOI:** 10.3390/membranes11010017

**Published:** 2020-12-24

**Authors:** Przemyslaw Jurczak, Emilia Sikorska, Paulina Czaplewska, Sylwia Rodziewicz-Motowidlo, Igor Zhukov, Aneta Szymanska

**Affiliations:** 1Faculty of Chemistry, University of Gdańsk, Wita Stwosza 63, 80-308 Gdańsk, Poland; emilia.sikorska@ug.edu.pl (E.S.); s.rodziewicz-motowidlo@ug.edu.pl (S.R.-M.); 2Intercollegiate Faculty of Biotechnology UG & MUG, University of Gdańsk, Gdańsk, Abrahama 58, 80-307 Gdańsk, Poland; paulina.czaplewska@ug.edu.pl; 3NanoBioMedical Centre, Adam Mickiewicz University, Wszechnicy Piastowskiej 3, 61-614 Poznań, Poland; 4Institute of Biochemistry and Biophysics, Polish Academy of Sciences, Adolfa Pawińskiego 5A, 02-106 Warszawa, Poland

**Keywords:** human cystatine C, NMR spectroscopy, micelle, DPC, SDS, dimerization, interactions

## Abstract

Human cystatin C (*h*CC), a member of the superfamily of papain-like cysteine protease inhibitors, is the most widespread cystatin in human body fluids. Physiologically active *h*CC is a monomer, which dimerization and oligomerization lead to the formation of the inactive, insoluble amyloid form of the protein, strictly associated with cerebral amyloid angiopathy, a severe state causing death among young patients. It is known, that biological membranes may accelerate the oligomerization processes of amyloidogenic proteins. Therefore, in this study, we describe an influence of membrane mimetic environment—mixed dodecylphosphocholine:sodium dodecyl sulfate (DPC:SDS) micelle (molar ratio 5:1)—on the effect of the *h*CC oligomerization. The *h*CC–micelle interactions were analyzed with size exclusion chromatography, circular dichroism, and nuclear magnetic resonance spectroscopy. The experiments were performed on the wild-type (WT) cystatin C, and two *h*CC variants—V57P and V57G. Collected experimental data were supplemented with molecular dynamic simulations, making it possible to highlight the binding interface and select the residues involved in interactions with the micelle. Obtained data shows that the mixed DPC:SDS micelle does not accelerate the oligomerization of protein and even reverses the *h*CC dimerization process.

## 1. Introduction

An amyloid is a name used for the description of a specific state of a protein (or peptide) in which its molecules are self-assembled (aggregated) into an insoluble form featuring a characteristic fiber-like structure. The phenomenon of protein aggregation is an important issue since it may lead to the loss of physiological activity or induction of non-physiological features of a protein, both of which may result in a disease state [1,2]. The diseases involving amyloid formation are generally called amyloidoses [3]. They include i.a. rheumatoid arthritis and Alzheimer’s, Parkinson’s, or Creutzfeldt–Jakob diseases [4,5]. In general, amyloidoses are severe states which cannot be cured—only symptomatic treatment is available. The problem is especially visible in the case of a subtype of amyloidoses affecting the central nervous system, where the mental impairment caused by the disease results in the need for constant medical attention and supervision over the patient who, in an advanced stage of the disease, cannot cope even with simple everyday actions. Due to the extension of lifespan, the nervous system-related amyloidoses are becoming increasingly common, causing an elevated socio-economic burden. As a result, a lot of effort is being focused on the studies related to the detailed description of protein oligomerization processes and possible factors and routes which could prevent them.

Current research indicates that biological membranes have a strong impact on amyloidogenic proteins and their oligomerization [6]. They may form the interface which accelerates the oligomerization process, thus promoting the formation of toxic, disease-causing, oligomeric forms. There are two potential oligomeric forms that are considered to be responsible for the toxic properties of the amyloidogenic proteins. One of them is an amyloidogenic fibril—an insoluble form, characteristic for all amyloidogenic diseases. Its toxic potential involves physical damage to cellular membranes [7]. Another oligomeric form of protein or peptide suspected to be the cause of the amyloidoses is an annular oligomer that may interact with cellular membranes forming channels, which disturb the membrane integrity [6]. While the formation of channels was proved for, e.g., amyloid A$\beta_{1-40}$ [8], α-synuclein, ABri, ADan, amylin, serum amyloid A [9], the formation of annular structures has been observed for many different amyloidogenic proteins and peptides (e.g., superoxide dismutase I, prion proteins, huntingtin, equine lysozyme, p53 tumor suppressor, islet amyloid polypeptide [10]), including human cystatin C (*h*CC) [11], which is the focus of this study.

According to the literature, there are two hypothetical pathways of oligomerization of amyloid proteins [6]. They both involve the formation of annular oligomers and lead to the aggregation of a protein into an insoluble fibril (Figure 1). One of the pathways assumes that the protein oligomerizes in the extracellular matrix and interacts with membranes only when the annular oligomer is formed. The other assumes that the membrane is the interface inducing and accelerating the oligomerization process. Both pathways are theoretically possible, but it was not yet unambiguously verified which one, if any, may be applied as a general model describing the oligomerization process of amyloidogenic proteins.

Human cystatin C (*h*CC) occurs naturally in the human organism, where it serves as a cysteine proteinase inhibitor [12,13]. It is common in all body fluids, and its physiological functions are linked mainly to maintaining the homeostasis of cathepsin-related intra- and extracellular enzymatic processes. Physiological activity of *h*CC is responsible for the proper functioning of an organism and supporting the immune system during bacterial infections [14]. On the other hand, the protein is also associated with some amyloidogenic diseases, where it co-accumulates with other amyloidogenic proteins [15,16,17]. Additionally, the occurrence of oligomers formed by the *h*CC L68Q mutant is directly correlated with the Icelandic type of hereditary cerebral amyloid antipathy (HCCAA), in which they accumulate in and damage brain arteries, causing strokes and death of patients at a young age [18]. The symptoms of the disease have already been described [19], but no cure has been developed yet.

Even though *h*CC dimerization and oligomerization processes are associated with a number of severe diseases [20], the routes leading to the formation of higher oligomers and fibrils of *h*CC have not yet been unambiguously described. The studies conducted earlier show that *h*CC undergoes transient changes from monomer to dimer during cellular trafficking and membrane crossing [21]. Therefore, we decided to focus on the studies on the influence of one of the proposed earlier modulators of amyloidogenic proteins’ oligomerization, i.e., the biological membranes, on the dimerization and oligomerization of the *h*CC, using the micellar membrane mimetics. The most commonly used micellar systems for membrane mimicking involve dodecylphosphocholine (DPC) and sodium dodecyl sulfate (SDS) surfactants [22,23,24,25,26]. The use of micellar membrane mimetics is especially common in NMR experiments due to significant methodological issues, causing the use of natural membranes to be tricky or even impossible [27]. In general, negatively charged SDS is used as a mimetic of prokaryotic cell membrane and zwitterionic DPC as a mimetic of eukaryotic cell membrane [27]. Mixed micelles are used to better mimic the natural properties of cellular membrane [28]. Therefore, to mimic the electrostatic properties of vertebrae plasma, which is characterized by a slight prevalence of the negative charge, we used a mixed micelle composed of DPC and SDS. The interactions between *h*CC and micellar membrane mimetics were described regarding the changes in the secondary structure and oligomeric state of the protein. For this purpose, size exclusion chromatography (SEC), circular dichroism (CD), and molecular dynamics (MD) simulations were applied. Additionally, using the nuclear magnetic resonance (NMR) techniques, we determined the regions of the *h*CC protein engaged in the interactions with the DPC:SDS mixed micelle.

## 2. Results

### 2.1. Selection of *h*CC Variants—What Are the Differences between *h*CC Variants’ Properties?

For the purpose of this study, the *h*CC variants were selected depending on their distinct properties in comparison to the wild-type (WT) protein, which, in physiological conditions, occurs in a monomeric form [29]. The *h*CC analogs with a mutation in position 57 were selected, since the valine (V57) is the most highly conserved residue in the structures of the whole cystatin family [30] and has a strained conformation what may be important for the biological activity of the protein [29]. Its exchange to other amino acid residues influences strongly the oligomeric state of the protein [31]. The *h*CC V57G occurs as a stable monomer in aqueous solution and, contrary to the WT protein, cannot be dimerized at increased temperatures and at acidic pH [32]. The V57P variant, on the other hand, possesses a stable dimeric structure and exists in the solution as the equilibrium mixture of monomer and dimer at ca. (circa) 1:5 ratio. It is also capable of further oligomerization or monomerization, depending on the environment. For the purpose of this work, the monomeric and dimeric WT protein and V57P variant were used during experiments visualizing the changes of the oligomeric state of the protein induced by micellar surfactants. The V57G stable monomer was used mainly during the experiments where the location of the regions of the protein involved in interactions with the surfactants was determined and dimerization would disturb the measurements or data analysis.

### 2.2. Size Exclusion Chromatography—How Does the *h*CC Monomer-Dimer Equilibrium Change in a Micellar Environment?

To verify the influence of the presence of micellar membrane mimetics on the dimerization and monomer-dimer equilibrium of *h*CC, size exclusion chromatography was applied. Initially, the *h*CC WT and its variants (V57G and V57P) were incubated for 24 h at 22 °C and 37 °C with micellar membrane mimetic DPC:SDS (5:1 molar ratio). In case of *h*CC WT (Figure 2a) and V57G (Figure 2b), no significant changes in monomer-dimer equilibrium were observed. In the case of partially dimeric *h*CC V57P variant (Figure 2c), the environment of the mixed micelle caused the change in the oligomerization state of the protein. Above the first critical micelle concentration (CMC) value 0.65 mM, the decrease of intensity of the signals representing dimeric *h*CC (retention time ca. 13.5 min) in the chromatogram was observed, with simultaneous increase of the intensity of the peaks with retention time ca. 16 min, associated with the monomeric *h*CC form (Figure 2c).

In order to verify if this interesting phenomenon is case-dependent only and limited to this specific *h*CC variant, the dimeric form of the WT protein was produced and subjected to a similar experiment. Two approaches were undertaken: using only dimeric *h*CC WT and a mixture of monomeric and dimeric *h*CC WT (1:1 molar ratio). The results of both experiments showed that, similarly to the *h*CC V57P variant, the monomer-dimer equilibrium shifts towards the monomer when the surfactant concentration increased above the first CMC value (Appendix A).

Further experiments involved incubation of *h*CC WT monomer, *h*CC WT dimer, and *h*CC V57P (dimer) in DPC surfactant solution (concentration of DPC equal to the concentration of DPC used earlier in DPC:SDS (5:1) mixed micelle). After incubation of *h*CC WT monomer (Appendix A), WT dimer (Appendix A), and *h*CC V57P (Appendix A) in DPC solution, no significant changes in monomer-dimer equilibrium were observed. Only incubation at increased temperature (37 °C) caused slight dimerization of the *h*CC WT monomer (Appendix A), suggesting that observed effect can be solely temperature-driven. After incubation of the *h*CC WT monomer, WT monomer/dimer mixture and V57P variant in SDS solution no significant changes to the *h*CC WT monomer (Appendix A) were observed, while the highest concentrations of SDS caused monomerization of *h*CC WT dimer (Appendix A) and *h*CC V57P (Appendix A). However, in comparison to the DPC:SDS mixed micelle environment, the SDS solution did not stabilize the monomerized form of *h*CC V57P. Significant lowering of the protein concentration, visualized by the decrease of the peak intensity, can be the result of protein degradation or aggregation at these specific conditions.

### 2.3. Circular Dichroism—Does a Micellar Environment Influence the Secondary Structure of *h*CC?

Circular dichroism spectroscopy was applied to verify the influence of the presence of micellar membrane mimetics on the changes of the secondary structure of *h*CC. The impact of the mixed micelle environment was studied first. For all analyzed proteins (*h*CC WT and its variants V57G and V57P), the interactions with the mixed micelle caused changes in the secondary structure, i.e., the increase of the content of the α-helix in the protein structure, as soon as the concentration of the surfactant increased above the first CMC value (CMC for DPC:SDS in PBS buffer equal 0.65 mM and 0.74 mM at 22 °C, and 37 °C, respectively). The increase of the strength of the molar ellipticity signal at the wavelengths ca. 208 nm and 222 nm was observed for *h*CC WT and both its variants (Figure 3a–c). An increase in the incubation temperature did not cause any significant changes in the results (Figure 3d).

The impact of separated components of the micelle on the protein was verified for *h*CC WT only. Since the changes caused by the mixed micelle were the same for all the studied *h*CC variants, the experiments for V57G and V57P were not performed.

As it appears, changes of the *h*CC WT protein structure (Figure 4a) caused by the DPC micelle (the highest concentration in the DPC dilution series 4.2 mM, equal to the concentrations of DPC in the mixed DPC:SDS micelle) were similar to those occurring as a result of interaction with the mixed micelle and manifested as an increase of the content of α-helices in the protein structure (an increase of the strength of the molar ellipticity signal at the wavelengths ca. 208 and 222 nm). The interactions with SDS (the highest concentration in the SDS dilution series 0.8 mM, equal to the concentrations of SDS in the mixed DPC:SDS micelle), on the other hand, gave slightly different results (Figure 4b). Only the highest concentration of SDS (0.8 mM) caused an increase of the content of an α-helix in the protein structure, even though significantly below the CMC value (1.1 mM at 22 °C). Lower concentration did not give a similar effect. Interestingly, the environment of 0.4 mM SDS caused degradation of the protein observed as a severe decrease of the intensity of the molar ellipticity signal.

### 2.4. Nuclear Magnetic Resonance—Which Fragments in the *h*CC Structure Bind to the Micelle?

The interaction between *h*CC V57G and DPC:SDS mixed micelle was monitored by the multidimensional NMR spectroscopy. The NMR experiments were collected for the *h*CC V57G variant, for which 3D high-resolution structure in solution was evaluated recently with NMR spectroscopy (pdb 6RPV) [32]. The experimental data acquired for *h*CC V57G in the presence of DPC-$d_{38}$:SDS-$d_{25}$ micelle suggests that the micelle caused structural alterations of the protein structure (Figure 5a). Detailed analysis of the cross-peaks positions shows shifts and changes in the intensity of signals for the residues Gly59, Ala103, and Thr111 (Figure 5a, insets). For several peaks, a dramatic increase of linewidth was observed, resulting in the disappearance of signals from ${}^{1}$H-${}^{15}$N HSQC spectrum, e.g., Val10, Val60, and Gln107 (Figure 5a, inset and Figure 5c).

The analysis of the chemical shift perturbations (CSP) reveals pronounced effects of the presence of DPC-$d_{38}$:SDS-$d_{25}$ micelle on the flexible loops linking anti-parallel β-strands of the *h*CC V57G. The interaction with the DPC:SDS micelle was detected for the ${}^{55}$QIGAGVNY${}^{62}$ and ${}^{104}$VPWQGT${}^{109}$ regions (Figure 5b), which are located in L1 and L2 flexible loops, respectively (Figure 6). The third structural fragment taking part in interaction with the micelle was detected within the N-terminus (${}^{10}$VGGPMD${}^{15}$, Figure 6). The addition of the micelle lead to the disappearance of the resonance for Val10 and changes in ${}^{1}$H chemical shifts for Met14 and Asp15 (Figure 5c). The occurrence of perturbations of ${}^{1}$H and ${}^{15}$N resonances confirms that the *h*CC V57G binds to the DPC:SDS mixed micelle using epitopes localized in the loops (L1 and L2) and N-terminus (Figure 6). Finally, the fact that the signals from Gly11 and Gly12 were also not detected, together with the neighboring Pro13, indicate that the possibility of unfolding of the short N-terminal β-sheet in the presence of the DPC:SDS micelle cannot be excluded. The comparison of ${}^{1}$H-${}^{13}$C HSQC spectra allowed us to analyze the changes in the position of signals representing methyl groups. Increased CSP values can be clearly seen for the majority of -CH_3_ groups in the presence of the DPC-$d_{38}$:SDS-$d_{25}$ micelle (Appendix A). The increased linewidth in the ${}^{1}$H dimension, observed for several cross peaks, suggests the existence of hydrophobic interactions between these residues and the micelle.

### 2.5. Molecular Dynamics—What Kind of Global and Local Structural Changes in the *h*CC Structure May Occur in a Virtual Micelle Environment?

The MD simulations were performed to visualize interactions between *h*CC (WT monomer, V57G monomer, WT dimer, and V57P dimer) and a DPC:SDS mixed micelle (molar ratio 5:1). The experiments conducted for three states of *h*CC V57G NMR structure (pdb 6RPV), named here V57G-1, V57G-2, and V57G-3, differed in position of the N-terminus (Appendix A). Based on the *h*CC V57G structures, three initial states of *h*CC WT (named here *h*CC-1, *h*CC-2, and *h*CC-3) (Appendix A) were obtained via single amino acid substitution (glycine in position 57 substituted with valine). The MD simulations of dimeric *h*CC WT (pdb 1G96) were performed based on the crystallographic structure of the protein. Similarly, as in case of the *h*CC monomers, *h*CC V57P was obtained via single amino acid substitution (valine in position 57 substituted with proline) in the structure of *h*CC WT dimer. To describe the extent of structural changes of the *h*CC molecules during the simulations, the root mean square distances (RMSD) was calculated, and, to verify which parts of the proteins interact with the micelle, and the solvent accessible surface area (SASA) values were calculated.

#### 2.5.1. DPC:SDS Mixed Micelle—hCC Monomer (WT and hCC V57G)

During the MD simulation the structure of hCC WT monomer stayed mostly unchanged (Figure 7 and Appendix A). The RMSD values, calculated for the overlaid (backbone atoms) structures of *h*CC WT before and after the MD simulation equaled 6.27 ± 2.71 Å (*h*CC-1), 2.85 ± 1.23 Å (*h*CC-2), and 4.57 ± 1.98 Å (*h*CC-3). The highest value was noted for hCC-1 state (showed here as a representative example (data for other states are presented in Appendix A). It results form the biggest change (compared to states *h*CC-2 and *h*CC-3) in the position of the N-terminal part of the molecule—flipped in opposite direction, expanding on the outside of the structure (Figure 7a). Noticeable movements were also observed in the region of appending structure (AS) of *h*CC-1 molecular state (Figure 7a), similarly to *h*CC-2 and *h*CC-3 states (Appendix A)). The core structure remained mostly unchanged; however, in the case of the WT molecule, some unfolding was observed in the α-helical part of *h*CC-2 state and β1-sheet part of *h*CC-2 and *h*CC-3 states. Additionally, some changes in the length of β2–β5-sheets were observed in the case of all states, but the unfolded sheets did not significantly change their position.

The interaction sites between the micelle and the monomeric *h*CC WT molecule determined during the MD simulations indicate that the micelle interacts mostly with N-terminal and L1, L2, and AS loop regions of the proteins (Figure 8, Appendix A). The results correlate with solvent accessible surface area calculations (Figure 7b, Appendix A). Fragments with the lowest solvent accessible surface area involve regions around N-terminus, α-helix, β2, β3, and β5 sheets, L1 and L2 loops, and parts of AS loop, with some variations depending on the NMR structure state.

Similar calculations were also performed for the V57G-1, V57G-2, and V57G-3 molecular states of *h*CC V57G variant. The obtained results indicate that the changes were very similar to those observed for the WT molecule. They involved some unfolding in the β1-sheet part of V57G-2 state of the structure and small changes in the length of β2−β5-sheets not accompanied by significant changes in their positions (observed in the case of all states (Appendix A). The greatest movements were again observed in the most flexible parts of the protein: N-terminus and the AS loop. The RMSD values, calculated for the overlaid (backbone atoms) structures of *h*CC V57G before and after the MD simulation equaled 4.00 ± 1.73 Å (V57G-1), 3.64 ± 1.58 Å (V57G-2), and 2.52 ± 1.09 Å (V57G-3). Smaller values, compared to *h*CC WT, confirm higher stability of the V57G protein variant. The protein–micelle interaction sites were analogous to the ones observed for WT protein and also showed some small variations between the analyzed models (Appendix A).

#### 2.5.2. DPC:SDS Mixed Micelle—*h*CC Dimer (WT and V57P)

During the simulation the structure of both dimers stayed mostly intact (Figure 9a and Figure 10a). Only some changes in the positioning of the flexible fragments (mostly within AS loop) occurred (Figure 9a). The RMSD values calculated for the overlaid (backbone atoms) dimeric structures before and after the MD simulation equaled 3.92 ± 1.70 Å for the dimeric *h*CC WT and 2.07 ± 0.90 Å for the V57P variant. High RMSD value for the WT protein is a result of a change of position of the WT dimer domains in relation to one another. Such movement was not observed in case of *h*CC V57P dimer.

The results of MD simulations indicate that the micelle interacts mostly with N-terminal, AS loop, and β−loop−β regions of the *h*CC WT and V57P dimers (Figure 9c and Figure 10c). The results correlate with solvent accessible surface area calculations. The fragments of both molecules with the lowest solvent accessible surface area involve the N-terminus, β−loop−β, and β5-sheet at the C-terminal end of the protein. Fragments of α1-helix, L2, and AS loops from both domains of the dimer also interact with the micelle. Selected fragments of the complexes formed by *h*CC dimers (WT and V57P) and the DPC:SDS micelle are highlighted in Figure 11 and Figure 12.

#### 2.5.3. Molecular Dynamics—Fluctuations

Further analysis of MD simulations data reveals relatively high structural displacements in the regions of the N-terminus, α-helix, and L1, L2, and AS loops for the *h*CC WT and V57G monomers. The Cα atom RMSF (root mean square fluctuations) averaged from three MD trajectories obtained for *h*CC-1—*h*CC-3 and V57G-1—V57G-3 are presented in Figure 13a. The data shows two important features: (1) as expected, the fluctuations in the β-strands are smaller than in the loop regions; and (2) for most of the residues, the RMSF are very similar (when comparing *h*CC WT and V57G). It should also be mentioned that, in the case of both, WT and V57G variants, increased fluctuations for α-helix can be observed, as well as slightly higher RMSF values for *h*CC WT compared to V57G.

Similarly. as in case of the monomers, relatively high structural displacements occurred in the regions of N-terminus, α-helix, AS, and L2 loops in the *h*CC WT and V57P dimers. The RMSF for backbone atom are presented in Figure 13b. The values for *h*CC WT (RMSF values between 4 and 8 Å) are considerably higher, compared to *h*CC V57P. This phenomenon occurs due to significant structural deviations of the entire dimer molecule during the MD simulation. They are mainly associated with the rearrangement of the dimer molecule where the individual domains move away significantly from each other (Appendix A). At the beginning of the MD simulation, the mass centers of domains in both dimer structures (*h*CC WT and V57P) were located at the distance of ca. 5 Å. After 25 ns of the simulation, the domains in the native molecule (*h*CC WT) began to move away from each other until they reach the distance of ca. 10 Å, at which they remain until the end of the simulation. In the case of the *h*CC V57P molecule, such changes were not observed. The increasing separation of the domains during the MD simulation results in bending of the dimer structure in the region of β-sheets linking the two domains. That is why the values of torsion angles were measured for the mutated residue 57th in monomeric and dimeric forms of *h*CC discussed in this study. Analysis of the backbone dihedral ψ and ϕ angles for the 57th residue enabled us to find a relationship between the type of amino acid residue in this position and the conformation of this residue and, as a consequence, the conformation of the L1 loop (in monomers) or β−loop−β structure (in dimers). The torsion angles for the residue 57th in the monomeric *h*CC WT possess wide distribution of ψ and ϕ dihedral angles and are located in two main clusters. The first cluster corresponds to an extended structure (ψ ca. −120÷−180 and 120÷180, ϕ ca. −120÷−180), the second to type I of the β-turn (ψ ca. 0÷−80 and ϕ ca. −120÷−180) (Figure 14a). The MD simulation of the *h*CC V57G variant resulted in only one, but very broad, cluster corresponding to the turn-like conformation of the 57th residue (Figure 14a), with dihedral angles ψ ca. −60÷180 ϕ ca. −120÷−180 and 130÷180. The residue 57th in the *h*CC V57G variant exhibits greater range of ϕ angles (compared to *h*CC WT), which is associated with higher conformational freedom of the Gly residue (compared to valine). In case of the dimers, the range of ψ and ϕ angles is limited to one cluster representative for the extended structure (Figure 14b). In the case of the V57P dimer, the residue 57th angles are in the classic (favored) βP region (ψ 120÷180, ϕ: −60÷−120; the range for Pro residues is greatly restricted because ϕ is limited by the cyclic side chain to the range −35 to −85). The *h*CC WT dimer angles are located in the allowed extended β-sheet structure region (ψ: 100÷180, ϕ: −100÷−180). The presence of the very stable conformation of the 57th residues in both dimers during the MD simulations stabilizes the extended β−loop−β structure in both dimers, but, in case of *h*CC WT, the structure is more extended.

## 3. Discussion

The description of the oligomerization process of *h*CC and factors influencing it is a potential starting point for the explanation of more advanced processes leading directly to the occurrence of the amyloidogenic diseases. Therefore, it is important to study how selected intra- and/or extracellular factors, such as, e.g., molecular membranes, modulate the oligomeric state of amyloidogenic proteins, such as *h*CC—do they prevent or promote the oligomerization? This knowledge may provide valuable information on the formation of toxic, disease-causing oligomers and fibrils and help to design new therapeutic strategies for amyloidoses. In this study CD spectroscopy, SEC chromatography, MD simulations, and NMR spectroscopy were applied to monitor the dimerization process of *h*CC protein and its interactions with micellar membrane mimetics.

### 3.1. Size Exclusion Chromatography—The DPC:SDS Micelle Reverses the *h*CC Dimerization Process

The SEC results show that the mixed micelle does not significantly influence the oligomeric state of monomeric form of WT *h*CC of its V57G variant. However, it has a strong impact on the dimeric variant V57P, where it reverses the dimerization process of the protein. In order to verify if this interesting phenomenon is case-dependent only and limited to this specific *h*CC variant, dimeric form of the WT protein was subjected to similar experiments as the monomer. What is interesting is that, during the incubation of the dimeric *h*CC WT in the micellar environment, several forms of this protein with retention time in the range between 13.5 min (dimeric form) and 16 min (monomeric form) were observed, depending on the concentration of the micellar solution. It is difficult to unambiguously associate the peaks with a certain form of the protein, especially taking into account the limitations of the gel filtration technique [33], in which not the exact size of the molecule but rather its hydrodynamic radius is the main discrimination factor [34]. In addition, the shape of the molecule matters, so the observed peaks may potentially illustrate either a monomer with slightly changed (loosened) conformation (the experimental conditions might have potentially cause partial unfolding of the protein), or more compact dimer.

Further experiments involving incubation of *h*CC WT monomer, *h*CC WT dimer, and *h*CC V57P (dimer) in DPC surfactant solution were performed to verify if the observed monomerization can be attributed to the properties of the mixed micelle environment and its electrostatic similarity to vertebrae plasma membrane [27,28] or, rather, results from the presence of a particular surfactant contained in the sample (DPC or SDS). The incubation of *h*CC WT monomer, WT dimer, and *h*CC V57P in DPC solution resulted in no significant changes in monomer-dimer equilibrium. Only incubation at increased temperature (37 °C) caused slight dimerization of the *h*CC WT monomer. This observation unambiguously excludes the effects of DPC surfactant as a sole origin of the observed changes and calls for further verification of the role of SDS.

As expected, the SDS micelle caused monomerization of the dimeric forms *h*CC dimers; however, in comparison to the DPC:SDS mixed micelle environment, the SDS solution did not stabilize the monomerized form of *h*CC V57P. SDS caused degradation of the protein in the highest concentrations (decrease of the peak intensity at the retention time 16 min and shift of the retention time to higher values) and a slight shift of monomer–dimer equilibrium to the dimer side (shift of the retention time from 16 min to ca. 13.5 min) when the surfactant concentration stayed below the CMC value.

### 3.2. Circular Dichroism—The Micelle Influences the Level of Order of the *h*CC Secondary Structure

The increase of order of the structure of *h*CC proteins observed in CD spectra occurred during its interaction with the DPC:SDS mixed micelle. It may be that the micelle separates fragments of the protein form in the aqueous environment, stabilizing them in the form of a highly ordered secondary structure. Similar effects were observed, e.g., for serum albumin A [35], amyloid A$\beta_{1-40}$ [36,37], and amyloid A$\beta_{1-42}$ [38], for which the presence of ionic surfactant caused the increase of order of the secondary structure of the molecule. The effect observed for *h*CC in the presence of the DPC micelle was similar and caused the increase of order in protein structure. A slightly different pattern was observed for SDS micelle, where only the highest concentration (0.8 mM) gave an effect similar to the one observed for the mixed micelle. Lower concentrations of SDS did not cause any significant changes in comparison to the control. Interestingly, the environment of 0.4 mM SDS caused degradation of the *h*CC protein observed as a severe decrease of the intensity of the molar ellipticity signal. This might have happened since SDS exhibits denaturing properties towards proteins and, at some concentrations, may cause protein degradation [35].

When combined, the CD and SEC data regarding interactions between *h*CC and the DPC:SDS mixed micelle indicate that the SDS surfactant is most likely the main component of the micelle responsible for most of the interactions causing the protein monomerization. The DPC surfactant is, on the other hand, an important factor stabilizing the monomer and preventing protein unfolding and degradation.

The monomerization process of the *h*CC V57P was unexpected and has no straightforward explanation. The literature data shows also that, at certain conditions, POPC membrane mimetics prevent formation of amyloid fibrils of A$\beta_{1-42}$ peptide [39]. It is possible that the biological membrane may also prevents oligomerization and fibrillization of other amyloidogenic proteins—including *h*CC. Therefore, it is possible that the membrane (or membrane mimetic) may cause partial monomerization of the protein by shifting the monomer-dimer equilibrium to the monomer side.

### 3.3. Nuclear Magnetic Resonance—Micelle Binds to the Flexible Parts of *h*CC Structure

The NMR experiments were performed for the *h*CC V57G variant due to the fact that we have previously successfully performed the NMR backbone sequential assignment for the protein (pdb 6RPV) [32]. The assignment for the WT *h*CC failed to be performed as a result of the partial dimerization process. The dimeric form of *h*CC is symmetrical, what caused difficulties in the assignment of the signals representing amino acids in different subunits of the dimer (problems with the signals’ separation occurred). Similar problems arose in the case of the dimeric variant of *h*CC (V57P). Therefore, the assignment of the *h*CC variants (besides V57G) needs further insight.

The initial ${}^{1}$H-${}^{15}$H HSQC spectrum recorded for the uniformly ${}^{13}$C${}^{15}$N-double labeled *h*CC V57G in the micellar environment demonstrated good dispersion of signals (Figure 5a). However, it was not possible to assign all the ${}^{1}$H and ${}^{15}$N chemical shifts based on the previous assignment performed for *h*CC V57G mutant in solution (bmrb 34399) [32]. The problems with acquiring the 3D NMR data (in spite of relatively good quality of ${}^{1}$H-${}^{15}$N HSQC spectrum) was most probably caused by the existence of strong interactions between *h*CC V57G and micelle, resulting in dramatic increase of molecular mass of the formed complex.

The analysis of epitopes taking part in interaction with the DPC:SDS micelle (L1 and L2 loops and N-terminal fragment) reveals that nearly all fragments involved in interaction are located in close proximity, forming an interface on 3D structure that can be bound by one micelle (Figure 6). Some residues exhibiting CSP were not considered as taking part in interaction due to very low intensity of signals in the NMR spectra and thus high measurement error (Figure 5b).

Detailed analysis of the sequence of the residues involved in interactions reveals that all of the fragments exhibit an isoelectric point at weak acidic pH. In particular, ${}^{19}$EEGVR${}^{24}$, ${}^{55}$QIGAGV${}^{60}$, and ${}^{103}$AVPWQGT${}^{109}$ fragments demonstrated theoretical pI equal 4.53, 5.52, and 5.57, respectively. Taking into account the low number of charged residues, we can speculate that majority of the interactions between *h*CC V57G and mixed DPC:SDS micelle occurred as a result of hydrophobic interactions between the protein and hydrophobic chains of the surfactants.

Interestingly, the determined *h*CC-micelle interaction sites overlap with the parts of inhibitor recognized by target proteases. The wedge-like shaped discontinuous epitope, consisting of N-terminus and fragments of both loops, is responsible for the interactions with papain-like proteases [40], whereas the AS structure is bound by asparaginyl endopeptidases from legumain-like family [41]. The significance of this observation awaits further studies; however, a significant (if any) impact on the physiological inhibitory activity of the cystatin C is rather not expected, especially taking into account very tight binding between the enzymes from the papain family.

### 3.4. Molecular Dynamics—Micelle Causes Structural Displacement in the Flexible Parts of *h*CC Structure

The MD results show that the *h*CC can interact with the mixed DPC:SDS micelle in both monomeric and dimeric forms. The micelle is attracted by the N-terminus, L1 and L2 loops, and partially β-sheets (which are connected by the loops) on one side and AS loop on the other side of the monomeric form of *h*CC protein (WT, V57G). In case of the dimeric form of *h*CC (WT, V57P), the micelle attracts the AS loops and the β−loop−β structure together with the L2 loops, located in close spatial proximity in space to β−loop−β fragment.

The comparison of RMSD values for different states of the *h*CC structures (*h*CC-1—*h*CC-3 and V57G-1—V57G-3) shows that, upon interaction with the micelle, the changes in the secondary structures occur in the flexible parts of the proteins, which are also mostly responsible for attracting the micelle. The parts of the protein that seem to contribute significantly to structural differences between free and micelle-bound molecules seem to be the N-terminus and the AS structure. Table 1 shows that, for the monomeric structures, the value of RMSD decreases significantly when the N-terminal part or AS-loop were omitted during the calculation. Only in the case of the *h*CC-2 and V57G-3 states of the NMR structure did the RMSD not change much when the N-terminal part was not involved in the calculation. This was due to the fact that, before the MD simulation, the N-termini of *h*CC-2 and V57G-3 were positioned in a similar manner as post-simulation. It should be stressed that, after the MD simulation, the position of the N-terminus was similar for all the states of the *h*CC NMR structures. It is tempting to hypothesize that this position may be adopted by the molecule in solution in the native fold. The lack of structural information for N-terminal part of any *h*CC variant analyzed so far (e.g., in the crystallographic studies) renders a critical evaluation of this hypothesis impossible.

The analysis of MD fluctuations led to similar conclusion. Once again, the highest changes were observed for the most flexible parts of the monomeric proteins (N-terminus and L1, L2, and AS loops). The fluctuations of the N-terminus occur due to naturally high flexibility of the region and its integration with the micelle. Slightly higher RMSD values observed for the whole *h*CC WT molecule indicate lower stability of the WT protein, compared to V57G variant. The increased fluctuations of an α-helical fragment were also observed. They are, however, associated with its movement, rather than greater flexibility.

In case of the *h*CC dimers the MD fluctuations show significant differences between the WT protein and its V57P variant. While the structure of *h*CC V57P variant stays mostly unchanged, the domains in WT molecule move away from each other. This phenomenon, resulting from the interaction of the protein with the micelle, may occur due to high flexibility of the dimeric *h*CC molecules in the hinge region (property found previously in 3D-domain swapped *h*CC crystal structures) [42]. This way, the protein may adapt its structure to the shape of the surrounding micelle during an interaction. Moreover, the analysis of dihedral angles of 57th residue in the studied *h*CC dimers shows that the extended β−loop−β structure is more favorable for *h*CC WT protein than for its V57P variant. This explains why, during MD simulation, the domains moved away from each other in the WT dimer and retained their position in the V57P dimer. On the basis of our previous data [30] and the literature [43], we found that the conformation of 57th residue could help in the explanation of the mode of action of the cystatin family molecules.

Finding a cure for amyloidogenic diseases is a herculean task. It seems that the best solution would be to use molecules which prevent or reverse the oligomerization process of the proteins. Recent studies allowed to discover such molecule for α-synuclein. A small organic compound, called SynuClean-D prevents aggregation, disrupts amyloid fibrils and blocks degeneration of dopaminergetic neurons characteristic for Parkinson’s disease [44]. Even though extensive studies need to be performed to prove the theory, it seems that the mixed DPC:SDS micelle (molar ratio 5:1) may exhibit properties which give a similar final outcome (prevention of fibril formation) for *h*CC as SynuClean-D for α-synuclein. And, if studies prove the hypothesis to be wrong, the micelle can be a good agent for stabilizing monomeric form of the *h*CC during in vitro studies.

## 4. Conclusions

This study was focused on the interactions between *h*CC WT and its variants with micellar membrane mimetics. The eukaryotic membrane mimetic (DPC) and prokaryotic membrane mimetic (SDS) were combined to form a mixed DPC:SDS micelle, with slight prevalence of a negative charge simulating vertebrae membrane better than DPC only. Literature suggests that the cellular membrane may serve as an interface promoting the oligomerization of the amyloidogenic proteins [6]. This, however, does not seem to be the case in regard to the interaction between *h*CC and the mixed DPC:SDS micelle (molar ratio 5:1). The monomeric *h*CC protein is not only stable in the micellar environment, but its structure is also more organized—the content of α-helix in the structure increases. The influence of the micellar environment on the dimeric form of *h*CC is also interesting. The micelle forces the dimer to monomerize. The results from molecular dynamics simulations and NMR titration experiments gave further information on the interactions between *h*CC and the mixed micelle. It seems that the interaction is strong, and the greater part of the protein is involved in it. The micelle binds not only flexible parts of the protein (N-terminus and loops) but also the sturdy β-sheets.

The obtained results show that the micellar membrane mimetics may reverse the process of oligomerization of amyloidogenic proteins, indicating that the idea of the biological membrane as an agent accelerating the process of oligomerization of amyloidogenic proteins needs further verification, at least in the case of *h*CC. However, to prove the statement, more extensive studies, involving more complex membrane mimetics and, in further perspective, natural membranes, are required.

## 5. Materials and Methods

### 5.1. Expression and Purification of Labeled and Unlabeled Proteins

The DNA of *h*CC variants V57G and V57P was obtained via site-directed mutagenesis as previously described [31]. Plasmid DNA (pHD313 vector [45]) containing: *h*CC gene coupled with signaling peptide derived from *E. coli* OmpA protein (causes the secretion of the protein into periplasmic space), temperature-sensitive λ cI 857 repressor, λ PR promoter, and ampicillin resistance gene, was transformed to and expressed in *E. coli* BL21(DE3) competent cells (Novagen) using standard LB medium and temperature-induced expression, according to the protocol described earlier [31].

The single (${}^{15}$N) and double (${}^{15}$N,${}^{13}$C) labeled *h*CC V57G variant was expressed as described earlier [32] with the use of single and double labeled minimal media containing ${}^{15}$NH${}_{4}$Cl and/or ${}^{13}$C-glucose (Sigma Aldrich, Germany). The isolation and purification of both labeled and unlabeled proteins was performed as described earlier in two steps with the use of ion exchange chromatography and size exclusion chromatography [31].

### 5.2. *h*CC WT Dimerization

*h*CC WT was dimerized using the acidic method proposed by Ekiel et al. [46]. The protein was incubated for 24 h in 25 mM sodium acetate buffer (pH 4.0) containing 150 mM NaCl and 1 mM benzamidine hydrochloride. Next, the dimer was separated from the monomer using size exclusion chromatography (column—Superdex 75 10/300 GL). The dimer was obtained with ca. 50% efficiency of the process.

### 5.3. Micelle Sample Preparation

Micelles form spontaneously above the critical micelle concentration (CMC; Appendix A); therefore, the preparation of the micelle solution does not involve any specific procedures. For the purpose of experiments described in this study, a micelle stock solution of DPC micelle, SDS micelle, and DPC:SDS mixed micelle (molar ratio 5:1) were prepared and used for the preparation of a dilution series. The concentrations of DPC and SDS in the mixed micelle solution equaled 8.4 mM and 1.6 mM, respectively (final concentration of the micellar solution equaled C${}_{DPC:SDS}$ = 10 mM). The DPC and SDS micelle solutions were used as controls for the mixed micelle solution. Therefore, the concentrations of the DPC and SDS in their solutions were equal to the respective concentrations of the surfactant in the mixed micelle solution (C${}_{DPC}$ = 8.4 mM and C${}_{SDS}$ = 1.6 mM).

### 5.4. Analytical Size Exclusion Chromatography of *h*CC

The *h*CC WT protein and two of its variants (V57G and V57P) at the concentration of 30 μM (in PBS) were incubated with DPC and SDS membrane mimetics or their mixture, at different mimetic concentrations (1:1, five step dilution series with the highest concentration of C${}_{DPC:SDS}$ = 5 mM, C${}_{DPC}$ = 4.2 mM or C${}_{SDS}$ = 0.8 mM depending on the mimetic used), for 24 h and at different temperatures (22 °C—room temperature, 37 °C—human body temperature). After incubation, 10 L of the mixture was applied on the gel filtration column (Superdex 75 PC 3.2/30, GE Healthcare Life Sciences, USA) and eluted with PBS buffer. The results were analyzed with Chromax 2007 (POL-LAB, Poland) and OriginPro 2018 software to verify if the *h*CC monomer-dimer equilibrium had changed.

### 5.5. Circular Dichroism Spectroscopy

The *h*CC WT protein and its two variants (V57G and V57P) were dissolved at the concentration of 0.15 mg/ml in PBS buffer. They were next incubated for 24 h at 22 °C (room temperature) and 37 °C with DPC and SDS membrane mimetics or their mixture at different mimetic concentrations (1:1 *v*/*v*, five step dilution series with the highest concentration of C${}_{DPC:SDS}$ = 5 mM, C${}_{DPC}$ = 4.2 mM or C${}_{SDS}$ = 0.8 mM, depending on the mimetic used). Before the experiment, the samples were spun at 10,000 g for 5 min to remove any insoluble particles from the solution. The CD spectra were registered with JASCO J-815 spectropolarimeter for supernatants at 22 °C in the UV range of 195–260 nm and analyzed with OriginPro 2018 (OriginLab Corporation, Northampton, MA, USA).

### 5.6. NMR Spectroscopy

The NMR samples consisted of 0.3 mM of uniformly ${}^{15}$N,${}^{13}$C-labeled *h*CC V57G dissolved in 50 mM phosphate buffer containing DPC-$d_{38}$:SDS-$d_{25}$ mixed micelle (molar ratio 5:1). The multidimensional NMR spectra were recorded at 298 K on Agilent DDR2 800 NMR spectrometer operating at 18.8 T (${}^{1}$H resonance frequency 799.786 MHz) and equipped with four channels, *z*-gradient Performa-IV unit, and ${}^{1}$H/${}^{13}$C/${}^{15}$N triple-resonance probe head. Acquired NMR data were referenced with respect to external sodium 2,2-dimethyl-2-silapentane-5-sulfonate (DSS). The ${}^{13}$C and ${}^{15}$N resonances were referenced in an indirect manner using coefficients Ξ = 0.251449530 and 0.101329118 for ${}^{13}$C and ${}^{15}$N, respectively [47]. Experimental data were processed with the NMRPipe [48] and analyzed with the Sparky [49] software. The chemical shift perturbations (CSP) for backbone ${}^{1}$H and ${}^{15}$N were calculated based on the following equation [50]: $\Delta \delta =sqrt[{\left(\Delta \delta H\right)}^{2}+0.154{\left(\Delta \delta N\right)}^{2}]$.

### 5.7. Molecular Dynamics

The MD simulations were performed using AMBER 16 package [51]. To create initial configurations, 70 DPC and 14 SDS molecules were randomly distributed around each protein using PACKMOL [52]. Double the number of surfactant molecules was engaged for dimer simulations. Each system was solvated in a truncated octahedron simulation cell filled with TP3 water, with the nearest distance between images of 8 Å. Sodium ions were added to neutralize the system. Then, each system was subjected to a 20,000-step energy minimization (steepest descent method). Afterwards, each system was heated from 0 to 300 K for 400 ps to avoid kinetic traps in local minima, followed by 80 ns MD simulations under periodic boundary conditions at 300 K, with a time step 2 fs and with isotropic pressure coupling. Long-range electrostatic interactions were evaluated by the particle Mesh Ewald (PME) summation. A cut-off of 10 Å was applied for non-bonded interactions. The SHAKE algorithm was used to constrain bonds involving hydrogen. Obtained data was analyzed with PyMOL software [53] and the solvent accessible surface area (SASA) and root mean square distance (for overlay of protein backbone) values were calculated for each *h*CC molecule with MOLMOL software [54].

## Figures and Tables

**Figure 1 membranes-11-00017-f001:**
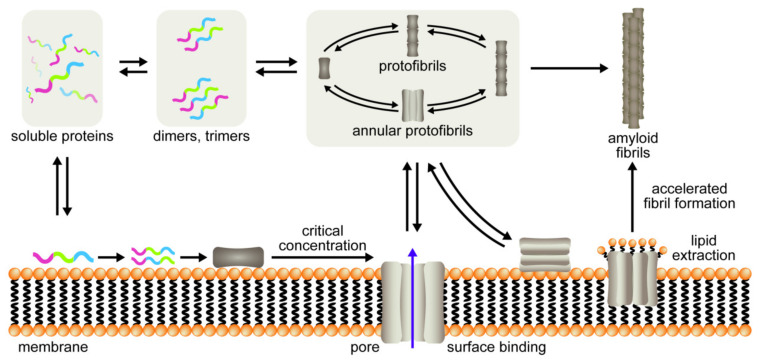
Pathways of amyloid formation; adapted with permission [6].

**Figure 2 membranes-11-00017-f002:**
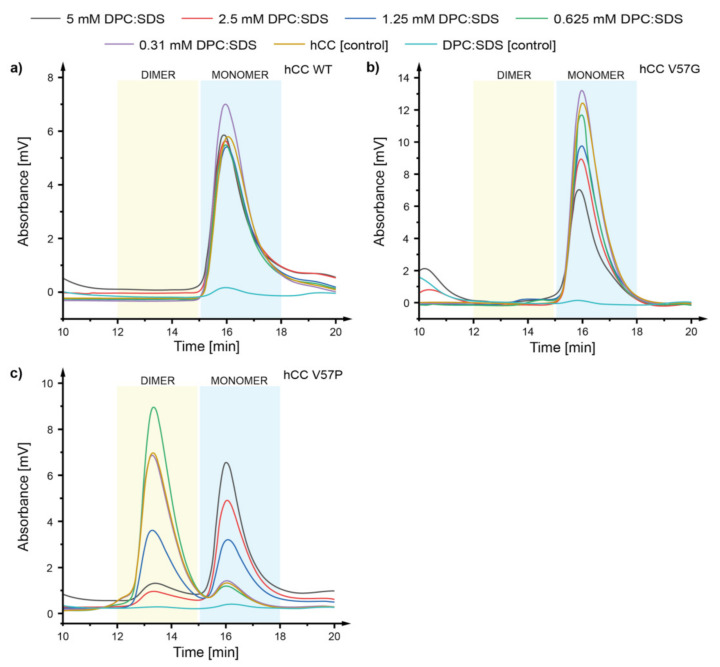
Chromatograms visualizing the oligomerization state of (**a**) human cystatin C (*h*CC) wild-type (WT) monomer, (**b**) *h*CC V57G, and (**c**) *h*CC V57P using the gel filtration chromatography after incubation at 22 °C for 24 h in the dodecylphosphocholine:sodium dodecyl sulfate (DPC:SDS) (5:1) mixed micelle solution; dimer retention time—ca. 13.5 min (yellow box), monomer retention time—ca. 16 min (blue box).

**Figure 3 membranes-11-00017-f003:**
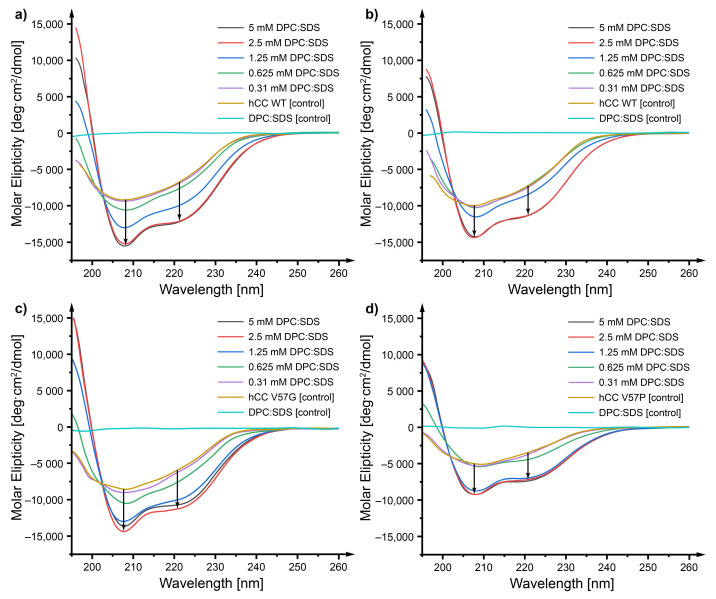
Circular dichroism (CD) spectra visualizing changes caused by the environment of DPC:SDS mixed micelle (5:1 molar ratio) in the secondary structure of (**a**) *h*CC WT, (**b**) *h*CC V57G, (**c**) *h*CC V57P after incubation for 24 h, and (**d**) *h*CC WT after incubation for 24 h at 37 °C.

**Figure 4 membranes-11-00017-f004:**
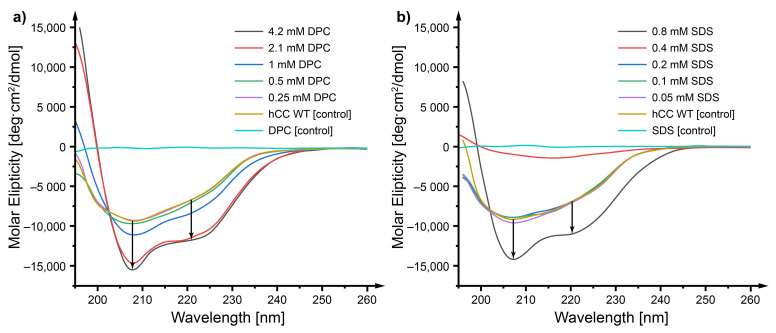
CD spectra visualizing changes of the secondary structure of the *h*CC WT influenced by the environment of biological mambrane mimetics: (**a**) DPC and (**b**) SDS for 24 h at 22 °C.

**Figure 5 membranes-11-00017-f005:**
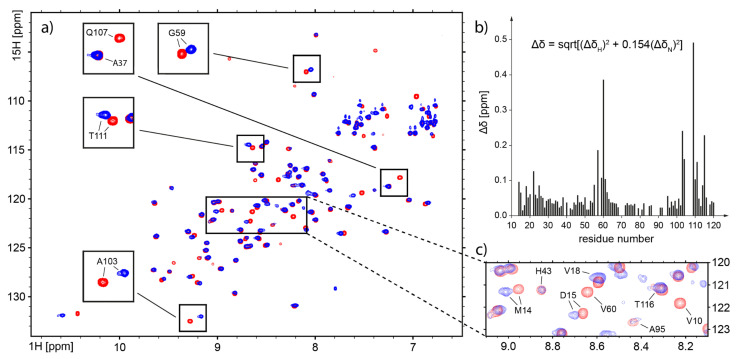
(**a**) Overlay of 2D ${}^{1}$H-${}^{15}$N HSQC spectra recorded for *h*CC (V57G) protein only (red) and with addition of DPC-$d_{38}$:SDS-$d_{25}$ micelle (blue); changes in the position of cross peaks observed for some residues zoomed in insets; (**b**) the chemical shift perturbations (CSP) plot for ${}^{1}$H and ${}^{15}$N chemical shifts in sequence-specific manner; (**c**) fragment of the overlay of ${}^{1}$H-${}^{15}$N HSQC spectra for *h*CC V57G, revealing changes in ${}^{10}$VGGPMD${}^{15}$ fragment of the protein, in the presence of DPC-$d_{38}$:SDS-$d_{25}$ micelle.

**Figure 6 membranes-11-00017-f006:**
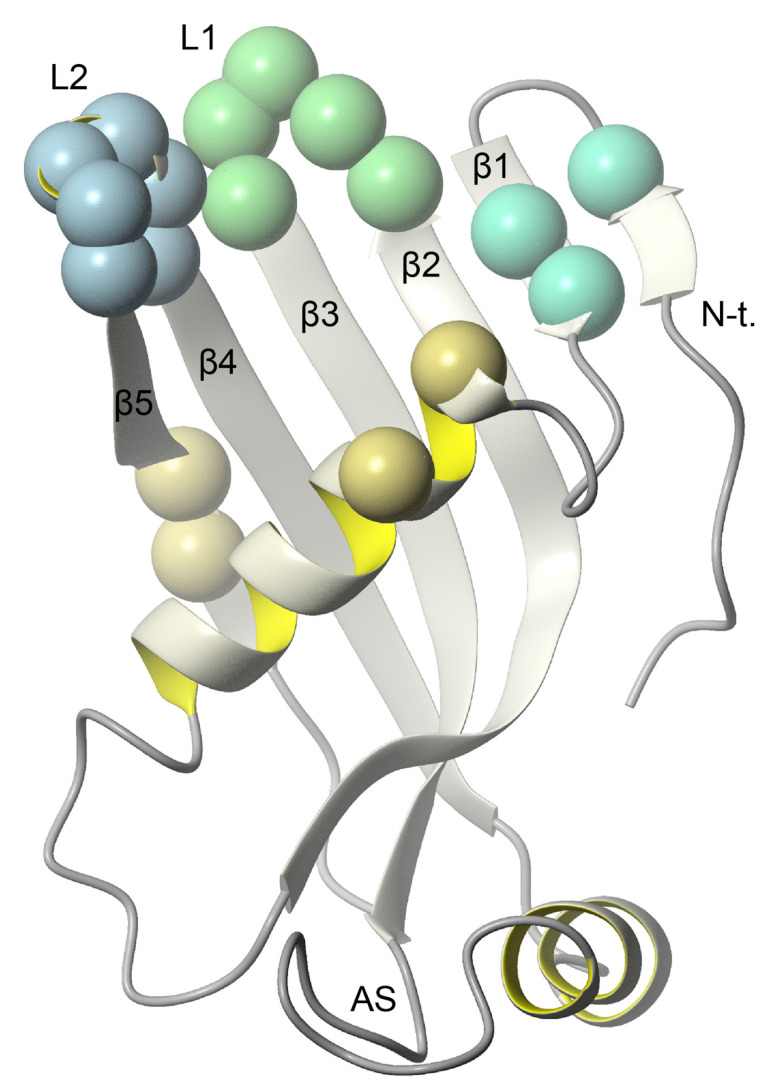
The high-resolution 3D structure of *h*CC V57G (pdb 6RPV); the amide groups exhibiting significant changes in the presence of DPC-$d_{38}$:SDS-$d_{25}$ micelle are highlighted in the form of marbles (blue, green, and turquoise).

**Figure 7 membranes-11-00017-f007:**
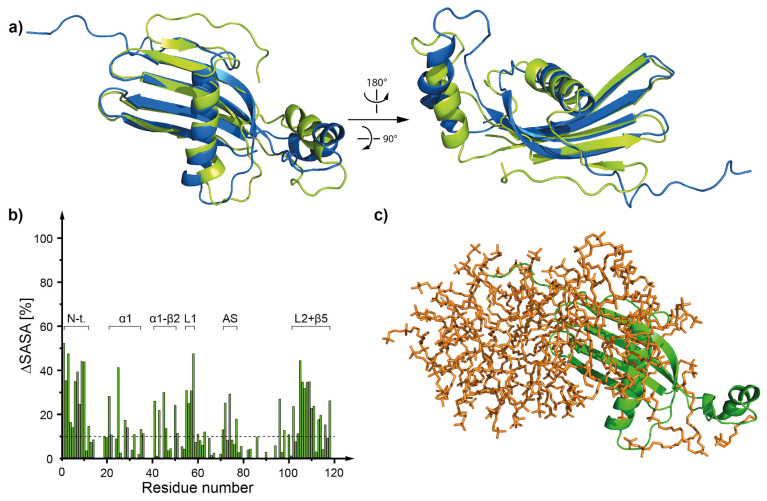
(**a**) The structure of hCC WT (hCC-1 state) monomer before (green) and after (blue) 80 ns of MD simulation of *h*CC-DPC:SDS mixed micelle interaction; (**b**) histogram visualizing the percentage of the decrease of the solvent accessible surface area occurring as a consequence of the interaction between *h*CC WT monomer and DPC:SDS mixed micelle; (**c**) a model of hCC WT monomer protein interacting with the DPC:SDS mixed micelle, corresponding to the histogram (structure after 80 ns of MD); ΔSASA (solvent accessible surface area) calculated as a difference between the SASA for the protein model without the micelle and protein model surrounded by the micelle.

**Figure 8 membranes-11-00017-f008:**
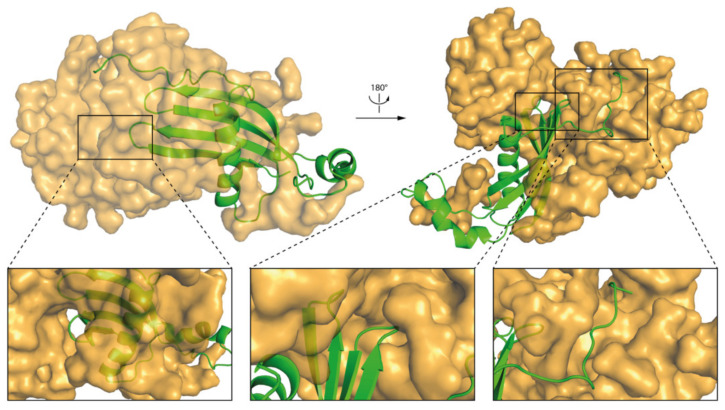
The structure of *h*CC WT (*h*CC-1 state) and DPC:SDS micelle complex after the MD simulation; insets present magnified fragments of the complex.

**Figure 9 membranes-11-00017-f009:**
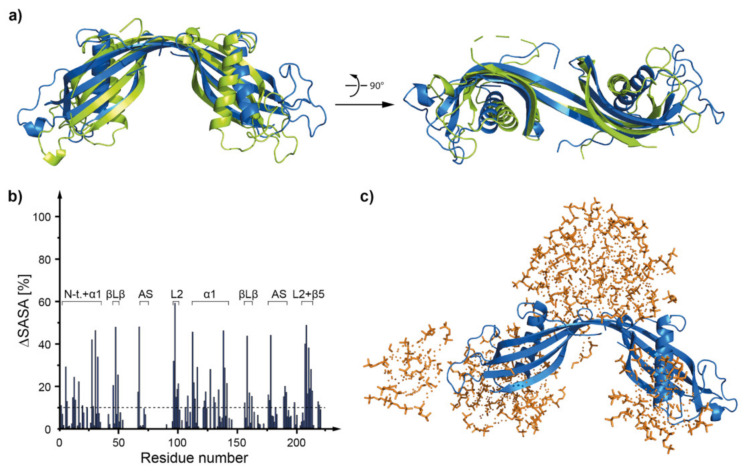
(**a**) The structure of *h*CC WT dimer before (green) and after (blue) 80 ns of MD simulation of *h*CC-DPC:SDS mixed micelle interaction; (**b**) histogram visualizing the percentage of the decrease of solvent accessible surface area occurring as a consequence of the interaction between *h*CC WT dimer and DPC:SDS mixed micelle; (**c**) a model of *h*CC WT dimer protein interacting with the DPC:SDS mixed micelle, corresponding to the histogram (structure after 80 ns of MD); ΔSASA calculated as a difference between the SASA for the protein model without the micelle and protein model surrounded by the micelle.

**Figure 10 membranes-11-00017-f010:**
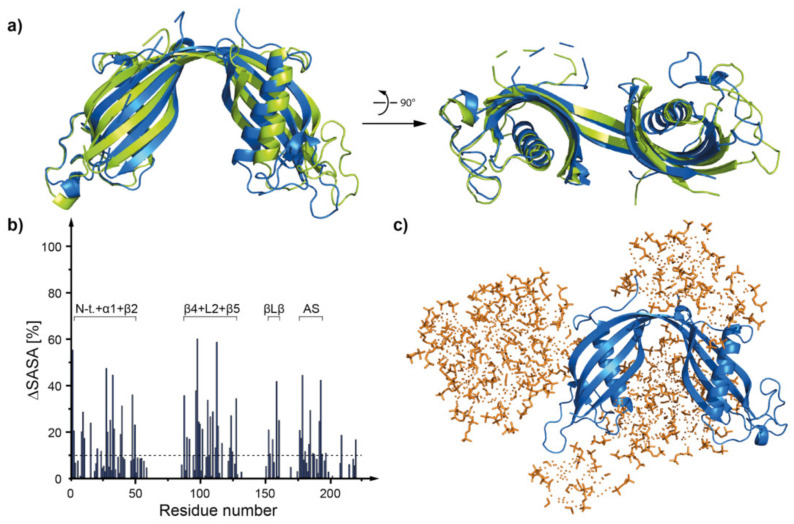
(**a**) The structure of *h*CC V57P before (green) and after (blue) 80 ns of MD simulation of *h*CC-DPC:SDS mixed micelle interaction; (**b**) histogram visualizing the percentage of the decrease of solvent accessible surface area occurring as a consequence of the interaction between *h*CC V57P and DPC:SDS mixed micelle; (**c**) a model of *h*CC V57P protein interacting with the DPC:SDS mixed micelle, corresponding to the histogram (structure after 80 ns of MD); ΔSASA calculated as a difference between the SASA for the protein model without the micelle and protein model surrounded by the micelle.

**Figure 11 membranes-11-00017-f011:**
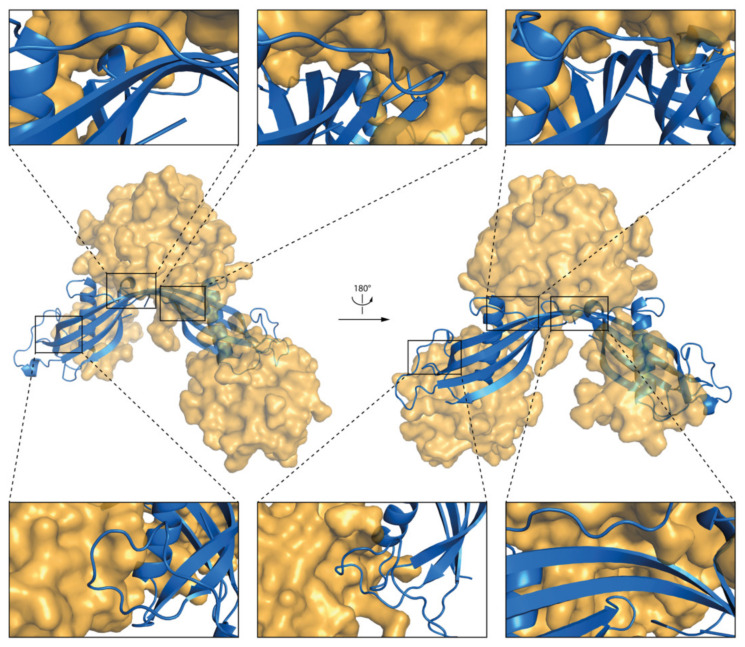
The structure of *h*CC WT dimer and DPC:SDS micelle complex after the MD simulation; insets present magnified fragments of the complex.

**Figure 12 membranes-11-00017-f012:**
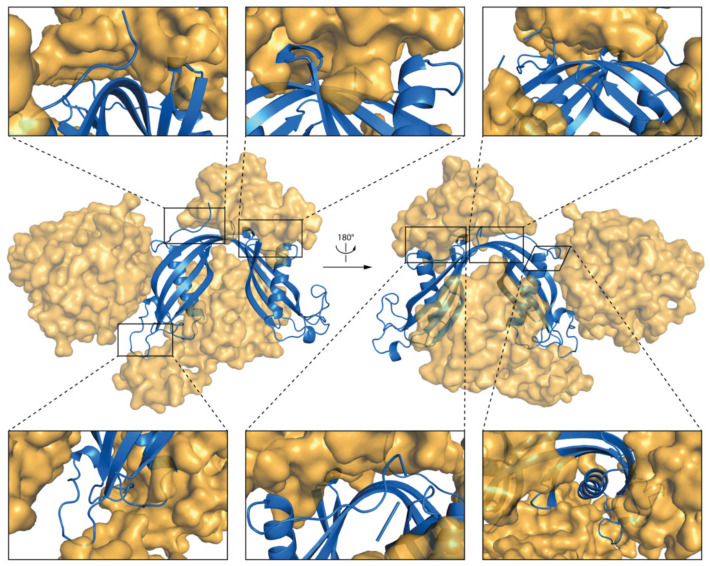
The structure of hCC V57P dimer and DPC:SDS micelle complex after the MD simulation; insets present magnified fragments of the complex.

**Figure 13 membranes-11-00017-f013:**
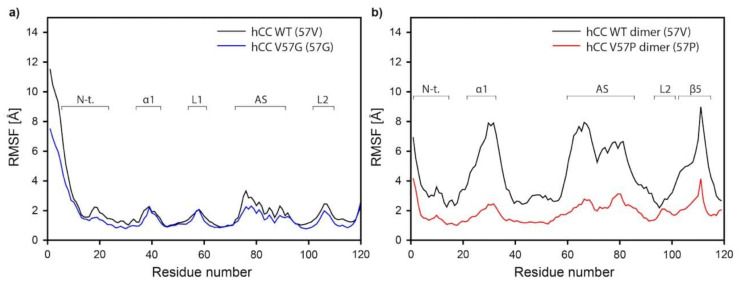
The backbone atoms root mean square fluctuations (RMSF) during the MD simulation; (**a**) *h*CC WT and *h*CC V57G monomers (averaged RMSF values from three MD trajectories) and (**b**) *h*CC WT and *h*CC V57P dimers; due to the symmetrical nature of *h*CC dimers, (**b**) shows fluctuations for half of the dimer structure.

**Figure 14 membranes-11-00017-f014:**
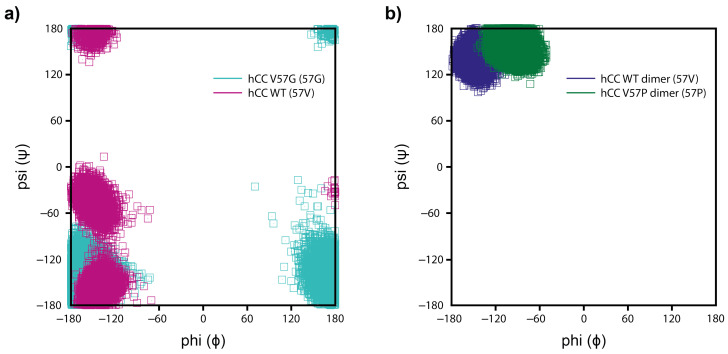
Scatter plots of the dihedral angles for the residue 57 (ϕ and ψ) in the (**a**) *h*CC WT and V57G monomers and (**b**) *h*CC WT and V57P dimers.

**Table 1 membranes-11-00017-t001:** Root mean square distances (RMSD) values extracted from different states of the *h*CC WT and V57G MD simulations.

Structure	Whole *h*CC	Protein (Monomer or Dimer) ^1^	Protein (Monomer) ^2^
*h*CC-1	6.27 ± 2.71 Å	2.05 ± 0.89 Å	1.38 ± 0.60 Å
*h*CC-2	2.85 ± 1.23 Å	2.72 ± 1.18 Å	2.49 ± 1.07 Å
*h*CC-3	4.57 ± 1.98 Å	2.74 ± 1.18 Å	1.73 ± 0.75 Å
V57G-1	4.00 ± 1.73 Å	2.56 ± 1.10 Å	1.48 ± 0.64 Å
V57G-2	3.64 ± 1.58 Å	1.84 ± 0.80 Å	1.56 ± 0.68 Å
V57G-3	2.52 ± 1.09 Å	2.48 ± 1.07 Å	1.53 ± 0.66 Å

^1^ without N-termini (residues 1–20); ^2^ without N-termini (residues 1–20) and appending structure (AS) loop (residues 73–94).

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
