# Peer review of "The Influence of the Mixed DPC:SDS Micelle on the Structure and Oligomerization Process of the Human Cystatin C"

_membranes, 2020, doi:10.3390/membranes11010017_

Round 1

Reviewer 1 Report

The authors presented biophysical and structural studies of an amyloidogenic protein - human cystatin C - in membrane mimicking environment. Specifically, using SEC, CD, and NMR methods as well molecular dynamics they compared the dimerization of cystatin C and its mutants interacting in different manner with negatively charged DPC/SDS micelles thereby modeling protein accumulation on membrane surface followed by a protein-lipid pore formation. This is an interesting, well conducted work concerning molecular mechanism of amyloidosis, a vital biological problem closely related with many human diseases. Nevertheless, there are several issues that the authors need to address before the manuscript is acceptable for publication in Molecules.

1) The eukaryotic membrane mimetic, zwitterionic DPC, would be better than charged DPC/SDS mixture for monitoring lipid-induced dimer-monomer transitions by SEC (Figures 2 and 3). Please comment.

2) Besides the 1H,15N-signal changes of amide groups (Figure 6), the availability of the recombinant 15N/13C-labeled cystatin C and its mutant forms allows direct monitoring of oligomerization-induced perturbations of the individual 1H,13C-signals of certain residues, e.g. using 1H/13C-HSCQC NMR-spectra.

Reviewer 2 Report

The authors presented a study of human cystatin C (hCC) and its potential oligomerization due the present of the micelle. In addition to hCC wildtype, two hCC variants were also considered to study the oligomerization/dimerization with micelle. Both experiments and simulations were used to investigate weather or not micelle can have an impact on hCC oligomerization. The conclusion is that a particular mixed micelle does not accelerate hCC oligomerization or reverse the dimerization. Overall, the manuscript shows solid investigation and a clear message.   Here is some comments for the authors:
  • The natural composition of micelle is not easy to artificially mimic in lab. Thus “the real micelle may or may not accelerate hCC oligomerization” remains unclear.
  • Does the size of micelle affect the overall observation from MD simulation? The MD simulation is expensive, and specially for explicit preotein/membrane simulation. It’s probably not a fair question, but is the model system used in MD simulation really large enough? 
  • Why does hCC WT monomer (Figure 14a) has less fluctuation than hCC WT dimer (Figure 14b)? Dimer is not expected to be more rigid than monomer?
  • In Figure 14,  y-label should be RMSF.

Reviewer 3 Report

There is an apparent reduction in the monomer peak intensity of V57G above 0.65 mM CMC (Fig 2b).  Where there large aggregates of V57G above 0.65 mM CMC?

Where there crosslinking of cysteine residues during WT dimerization?

Fig 3-  is not convincing and maybe deleted.  There is a tailing effect, and distinct peaks are not seen as in Figure 2.

Define “ca.” at the first usage.

The conclusions are based on monomer-dimer equilibrium.  Some of the figures suggest the possible presence of large oligomers or destabilized aggregates, which could accelerate amyloid fibrillogenesis.
